# Training Packages and Patient Management Tools for Healthcare Staff Working with Small, Nutritionally At-Risk Infants Aged under 6 Months: A Mixed-Methods Study [note 1]

**DOI:** 10.3390/children10091496

**Published:** 2023-09-01

**Authors:** Ramona Engler, Marie McGrath, Marko Kerac

**Affiliations:** 1London School of Hygiene and Tropical Medicine, Keppel Street, London WC1E 7HT, UK; ramona.engler1@alumni.lshtm.ac.uk (R.E.); marie.mcgrath@lshtm.ac.uk (M.M.); 2Emergency Nutrition Network, Oxford, OX5 2DN, UK

**Keywords:** infant, malnutrition, training, management, care pathway, nutrition, child health, MAMI, wasting, underweight

## Abstract

Skilled staff are essential for successfully managing child malnutrition, especially when dealing with small, nutritionally at-risk infants aged under 6 months. Training and patient management tools provide healthcare workers with core knowledge, skills, and support. To inform more effective future approaches and support the rollout of new WHO wasting prevention/treatment guidelines, we aimed to map and understand globally available training and patient management tools. In a mixed-methods study, we searched the literature to identify different training packages and management tools and conducted semi-structured key informant interviews with staff working in a variety of internationally organizations and settings. Data were analyzed using a phenomenological approach. We found 14 different training packages targeting different settings, including inpatient, outpatient, and humanitarian contexts. Curricula varied, especially around breastfeeding and maternal assessment, mental health, and nutritional status. Key informants (n = 9) reported gaps regarding counseling skills, time for practice, and how to effectively provide mental health counseling. Training delivery was highly dependent on resources and setting. In conclusion, future training should consider setting-specific needs, opportunities, and limitations. Emphasis on breastfeeding-support skills, mental health support, and counseling skills is needed. Enhanced practical sessions, refresher trainings, and ongoing monitoring and support are vital to ensure sustained high-quality services.

## 1. Introduction

Millions of infants aged under 6 months (infants u6m) worldwide are small or nutritionally at-risk [1]: an estimated 20.1% (23.8 million globally) are underweight; 21.3% (24.5 million) are wasted; 17.6% (21.5 million) are stunted [2]. The recently released July 2023 WHO Guidelines on prevention and management of wasting highlight this group as being “at risk of poor growth and development” and there is an entire new guideline section devoted to their care as a critical preventive action [3]. Risks of failing to provide effective, well-delivered interventions include: **High risk of death in the short term**: an anthropometric deficit has long been recognized as a major risk factor for mortality [4,5,6].**High risk of morbidity including poor development:** there are not only short-term risks of this [7], but also long-term cognitive and behavioral sequelae [8].**Long-term risks of cardiometabolic and other non-communicable disease** [9,10]: these are explained by the ‘DOHaD’ (Developmental Origins of Health and Disease) [11] and ‘capacity-load’ models [12].

As outlined by the UNICEF conceptual framework [13], malnutrition in all its varied manifestations is influenced by a multitude of underlying factors (note that our term ‘small, nutritionally at-risk’ covers all the major forms of malnutrition and includes infants with low weight-for-age, low weight-for-length, low mid–upper arm circumference, low length-for-age, and those who were born premature, small-for-gestational-age, and/or with low birth weight [14]). Given the many possible causes, there is a need to consider which applies to an individual infant/mother pair. For example, there are both immediate causes (e.g., an acute illness such as an episode of diarrhea; an infant who has had recent surgery for a congenital anomaly) and more distal/upstream causes (e.g., poverty; poor environment; underlying disability that indirectly leads to undernutrition.) Factors underlying or contributing to vulnerability in infants u6 months are particularly varied, with three main categories of risk factors: maternal, household, and infant [15]. For example, statistically significant and biologically plausible associations have been found between infant u6m wasting and: poverty, maternal nutrition, mothers’ education, mothers’ disempowerment, being born small, delayed introduction of breastfeeding, prelacteal feed, and recent illness with diarrhea [15]. Other evidence shows strong associations between maternal mental health and the infant’s nutritional status and health [16,17]. Having such a range of possible underlying causes makes effective management of small, nutritionally at-risk infants u6m challenging, since staff must have the skills and experience to both identify and, where possible, address the issues affecting a particular infant/mother pair.

Major positive developments for small and nutritionally at-risk infants u6m are the newly released July 2023 WHO malnutrition guidelines, which have an entire section, one of four major sections, devoted to their care [3]. However, many of the recommendations therein are based on low certainty evidence. Relatively little new research focused specifically on this sub-population has taken place since the group were first highlighted in the 2013 WHO malnutrition guidelines [18]. It is hence vital to look at, learn from, and work with other programs and initiatives working on closely related areas of infant and maternal health, such as small vulnerable newborns [19].

As national and international policy makers and programmers consider and look to roll out and translate the new 2023 WHO guidelines from global-level concepts and recommendations to practical actions made at country and local levels, another challenge is to ensure that the front-line implementation is as smooth and effective as possible. A number of recommendations made regarding infants u6m in the 2013 WHO guidelines have still not filtered down to national policy [20]. For example, WHO recommended—and still recommends—outpatient-based care for malnourished infants u6m who are clinically stable. Yet almost all current country-level guidelines retain the old-style inpatient-only care model. One problem with such inpatient-based strategies, is that they rarely succeed in practice [21]. A likely reason for the lack of outpatient services is the previously noted issue that effective care for infants u6m requires a broad skill set. If policy makers are not confident that front-line community-based healthcare workers have these, they will be cautious in recommending a new outpatient-based care approach. Policies to disseminate breastfeeding support and relaxation units are important, as is the need for training in breastfeeding support at all stage of healthcare workers’ careers (including during basic training and postgraduate on-the-job training). Common skill gaps for healthcare workers dealing with infants u6m include: breastfeeding support [22]; maternal health and mental health assessment [23]; assessment and care for infant problems [23]. All of these require not only robust technical skills but also ‘soft’ skills such as counseling and interpersonal skills.

Considering all of these challenges in managing small vulnerable infants u6m, the need for well-trained staff who are both empowered and enabled to deliver caring, quality support as effectively as possible is clear. However, since this is a relatively underappreciated nutrition/health topic, specific training materials are not yet widespread. To help develop these in future, there is a need to learn from what has been done to date, including in other closely related programs/services. These related training tools and patient support materials can then be adapted and built on rather than developing new materials from scratch. Thus, the aim of this study is:To improve the future management of small and nutritionally at-risk infants and their mothers via an improved understanding of currently available training packages and patient management tools.

Objectives are:To identify how many training programs and tools are currently available and their differences.To identify perceived issues and gaps in the practical application of training.To identify how training can be improved to ensure optimal delivery.

## 2. Methods

### 2.1. Study Design

The study combines two methods; firstly, a scoping review of the literature to identify different training packages relevant to staff managing at-risk infants u6m and their mothers; secondly, a qualitative study involving key informant interviews to provide an insight into how training is used and perceived, possible issues, and room for improvement. The study initially began as an MSc summer project at the London School of Hygiene and Tropical Medicine: the full text of this thesis is available online [24].

### 2.2. Review of the Literature

The focused literature review included any training currently available that had a broad overall aim of facilitating staff working with vulnerable infants and their mothers. Search terms included: “training tools”; “training manuals”; “training packages”; “guidelines for training staff working with infants under six months”; “guidelines for staff working with infants u6m”. Training aimed at different levels of staff, ranging from management staff to front-line healthcare workers, was included. Operational guidelines were excluded. 

### 2.3. Qualitative Research

The theoretical framework of the qualitative research was based on a phenomenological approach. To ensure completeness of the research methods description, the ‘Consolidated Criteria for Reporting Qualitative Research’ (COREQ) checklist was used [25].

#### 2.3.1. Participant Selection

A purposive sampling method was chosen, as it allowed for the sampling of participants who had relevant experience with the phenomena being investigated [26]. Initial contacts were from our personal professional networks via the MAMI Global Network: those managing programs and/or working directly with small, nutritionally at-risk infants u6m [14]. Invites were sent via e-mail and included information about participation in the study and a consent form. 

From these contacts a purposive nonprobability sample was drawn until the point of data saturation was achieved [27]. A sample of at least 10 interviewees was considered appropriate, aiming at a balanced sample of researchers working for different organizations. In total, 17 invitations were sent, and 15 individuals responded. Due to challenges in arranging interviews and time constraints a final number of nine participants was included in the study.

#### 2.3.2. Data Collection

Data were collected and analyzed by the lead researcher (R.E.) who, at the time of the study, was an MSc student in Nutrition for Global Health at the London School of Hygiene and Tropical Medicine. Prior to the research commencement there was no relationship between the researcher and participants. Interviewees were informed about the researcher’s background, goals, and interest at the beginning of each interview.

For primary data collection, a semi-structured interview outline was developed. Questions that would provide an insight into topics relevant to the objectives were chosen. A draft interview outline was sent to participants to provide an overview and guidance on the topics covered. Interviews aimed to collect data on the experience of using training targeted at staff working with at-risk infants u6m, covering three main areas corresponding to the main study objectives:

Awareness of different training available and experience of its use (*Objective 1*).

Identification of perceived issues and gaps (*Objective 2*).

Identification of how training could be improved (*Objective 3*).

The outline covered seven questions, ranging from general information on previous experience with at-risk infants u6m to questions about different training, including strengths and limitations, as well as what an ideal training would look like. The outline was used as a rough guide; however, participants were able to share experiences outside these questions that seemed relevant to them. Interviews were held from the 1st to the 10th of August 2019. 

At the beginning of each call, the researcher introduced the project with special emphasis on the goals and objectives to ensure that participants were familiar with the research and able to share knowledge and experience relevant to the research question. Furthermore, it was ensured that participants felt confident to share their knowledge on the topic. The duration of the interviews ranged from 20 min to one hour.

Each interview was only held once and during the conversation nobody else other than the researcher and interviewee was present.

Interviews were held and recorded using Skype (version 8.46.0.60, 2019 Skype and Microsoft) on the researcher’s private laptop. The audio recordings were stored in a password encrypted folder to ensure safe storage and confidentiality. 

Due to time limitations, it was not possible to return the written transcripts to participants for proofreading and agreement with the content. However, each interviewee was informed that this would not be possible and agreed to the use of their provided information.

#### 2.3.3. Data Management and Analysis


**
*Management*
**


Data were handled only by the main researcher. No field notes were taken during the interviews. In order to guarantee the participants’ confidentiality, names were replaced with participant IDs. Organizational names were also not recorded.

For faster transcription, the software Express Scribe Pro (NCH Software v.5.55) was used. Interviews were transcribed completely. Familiarization with the data took place during the transcription process and for further management of the analysis NVivo (QSR International, version 12, 2018) was used.


**
*Analysis*
**


Data were analyzed using a phenomenological approach to provide a complete and full insight into the participants’ experience. Findings could arise rather than being imposed by the researcher and the questions formulated in the interview guide served as a tool to achieve a deeper insight into the participants’ lived experience. The description of the findings was kept as close to the raw data as possible. This was achieved by careful familiarization with the data and by not adding to or deleting any meaning from the initial content of the interviews. The investigator aimed to minimize bias by being mindful of not being influenced by their own opinions and expectations when coding the themes.


**
*Derivation of Themes*
**


To ensure that the research objectives were addressed during analysis, the interview questions were categorized into three major themes: variety of training identified, differences in training, and areas of improvement. Due to the choice of a phenomenological research approach, themes were derived from the data.

#### 2.3.4. Ethical Considerations

The study protocol was reviewed and received ethical approval by the LSHTM MSc Research Ethics Committee, with the reference number: 17369. A consent form was signed by each participant before each interview. Participants’ anonymity was ensured by using participant IDs rather than names or other identifiable details.

## 3. Results

### 3.1. Identification of Training Programs and Patient Support Tools

The focused literature review found 14 training programs/patient management tools relevant to our target population of small, nutritionally at-risk infants u6m and their mothers. There was significant overlap between training resources and patient management tools and the two are therefore considered together. Table 1 shows the packages and tools identified and presents an overview of the different topics covered by each.

Appendix A show further details of each tool/support package.

Most of the training programs and support tools are directed at trainers (A, B, G, J) or directly at health staff, including community health workers (CHW), health workers (HW), nurses, counselors, and nutrition workers at the primary referral level (C, D, F, G, I). Some are aimed at different levels. For example, IMCI addresses staff such as doctors, nurses, and health workers directly involved with patient management, while the ‘Infant and Young Child Feeding Counselling’ package is the only training program specifically designed to facilitate illiterate participants and/or staff with a lower education background. ‘Getting to Know Cerebral Palsy’ and the ‘Feeding and Positioning Manual’ are the only two training programs that are directly aimed at caregivers. Three training programs target emergency relief staff, ranging from trainers (A, K, L), technical staff, and coordinators (K), to psychologists, lactation counselors, to midwives and nurses (L). 

All training programs aim to address mothers and infants, although cMAMI is the only one specifically targeted at small and nutritionally at-risk infants u6m. Others cover a broader age range, from 0–24 and the IYCF-E Toolkit even covers up to 60 months. ‘Thinking Healthy’ is targeted at mothers. (N.B., since this project was undertaken there has been a further update of cMAMI, the version 3 MAMI care pathway [42]).

IMCI is the only training program that is focused on inpatient care, although it was extended to also include the community. Four training programs are targeted at emergency settings (A, F, K, L) and one is specifically for low resource settings (N) where no technical devices are available. Two training programs do not specify a setting at all (D, M).

Four training programs did not have a specification on duration (A, F, I, M). One is a three-day course (G), three outlined that it was dependent on the setting (J, K, L), and three are set to be five-day training courses (C, D, H, N). IMCI intended to be undertaken over 11 days.

The main differences observed in the curricula are regarding assessment, breastfeeding, assessment of the mother’s mental health, and counseling skills. While most training programs, except G, H, M, and N, include an assessment of the infant’s feeding, only four (E, I, L, M) cover anthropometric and nutritional assessment. All training programs except ‘Thinking Healthy’ cover BF, however, some training programs go into more detail regarding BF assessment, issues, and counseling. Only training programs B, F, I, K, and L address relactation and provide information on how to help a mother to re-establish BF. Only H, I, K, L, and N included an assessment of the mother’s mental health. All training programs, except, A, D, G, and J, cover basic recommendations for maternal nutrition and physical assessment. While some training programs only provide basic recommendations for maternal nutrition (B, E, M, N) others (C, F, H, I, K, L) go into more detail about topics such as how to break the cycle of malnutrition in mothers, specific food requirements for health during pregnancy, lactation and general maternal care, and physical and reproductive health. Counseling skills were not included in the curriculum of training programs A, I, and M. All training programs focused on emergencies (A, F, K, L) and D covered the management of artificial feeding and donations. However, only F, K, and L addressed emergency preparedness. How to integrate training into other nutrition programs such as CMAM was only covered by training programs E, I, K, and L. Food hygiene was only included in the curriculum of four training programs (B, F, K, L)

### 3.2. Interviews

Key informant interviews served to distinguish between theoretical key features of training programs/tools outlined during the review of the literature and their actual use in practice.

#### 3.2.1. Participant Profile

In total, a sample of nine participants was included, working for nine different organizations and having experience across three different setting types. This is shown in Table 2.

#### 3.2.2. Training and Management Tool Characteristics According to Key Informants

During the interviews, participants identified the logistics of training programs they had used in the past as well as a rough outline of the curriculum and guidance they are based on. The results are displayed in Table 3. A wide variety of timings and approaches are observed, though a common feature is a relatively short training duration, especially given how much information there is to learn and understand.

#### 3.2.3. Themes Arising

Figure 1 summarizes key themes arising in the key informant interviews.

#### 3.2.4. Skills and Needs


**
*Counseling skills*
**


Interviewees considered knowledge of the topic as a very important need when it came to working with at-risk infants and their mothers. However, even greater emphasis was given to the importance of counseling skills amongst most participants interviewed, regardless of the training program. However, interviewees that put strong emphasis on counseling skills often had a community or emergency background.


*“Because it is actually really important to get the technical knowledge for the staff but actually if they don’t know how to do counselling and I have seen it in so many different countries, then it is not gonna be successful.”*

*ID3*


It was mentioned that, in public health, often too much focus is on general recommendations, anthropometrics, and feeding but participants felt that the “bigger picture” (ID3) needs to be considered and to “*follow a more holistic approach*” (ID3) would be to provide good counseling and support. This includes active listening skills, to identify what difficulties the mother might face regarding breastfeeding, encouraging her, and promoting active behavioral change to successfully achieve sustainable changes in infant feeding practices.


*“…they know it is the best thing for the baby, but they have too many other problems and so listening to those other problems I think is important.”*

*ID9*


Furthermore, engaging with people’s attitudes and beliefs, as well as storytelling, was thought to be an important part of good counseling skills. Compared to our Western culture, in many settings people are much more aligned with their beliefs and see them as part of their daily lives. Especially in settings in which people are confronted with life and death, engaging and connecting with people on this level is key to ensure engagement in the project. 


*“I mean, skills are one thing but really engaging with people’s attitudes and values (…) think we neglect that part (...) I mean for example in lot of what we are talking about are differences in life and death, so it is about how do you take the responsibility for that role and what does that mean to you as a person, why are you motivated to do this job, that kind of stuff.”*

*ID2*



**
*In-Depth Knowledge of the Topic*
**


Most respondents agreed that in-depth knowledge is an important skill for trainers and front line staff. 


*“…especially when it comes to helping with breastfeeding which is a lot of (...) in the dark, you need to understand malnutrition and you need to understand about breastfeeding and also a little bit about how you can spot a preterm baby. I mean a lot of those babies are actually not malnourished, but they are born preterm unless you know the gestational age of the baby then you label it as malnourished.”*

*ID2*


#### 3.2.5. Perceived Issues and Gaps

Issues and gaps also varied depending on the setting but were similar in some cases. Different tools had different issues and gaps and thus were differentiated in cMAMI, Save the Children training programs, and training programs used in inpatient care.


**
*Complicated Toolkit*
**


Especially with the cMAMI toolkit, participants found that the second version was more complicated compared to the first version. It was found to be “*a lot of material and the team felt quite overwhelmed with the materials*.” (ID 8). 

In training settings in which time is limited, staff often did not have the time to familiarize themselves with the tool and did not feel confident with its application. 


*“There was a little bit of practical on it, but with the extent of this one-day orientation the team does not feel confident to then cascade it down when I was checking last week. And this is from some who are experienced in IYCF and acute malnutrition.”*

*ID8*



**
*Assessment Criteria*
**


Participants (speaking in interviews before WHO 2023 guidelines, in which these were much more clearly specified) agreed that there was a lack of fixed, well-defined assessment criteria, which are important to identify admission and discharge cut offs. It was outlined that changes in assessment criteria, from weight for height to MUAC or weight for age, still create confusion as to which measure should be used. This not only impacts the implementation amongst organizations but also leads to governments not taking the tool on board.


*“…it is still heavily NGO-supported because we made it a bit too complicated and if we keep changing the admission and discharge cut offs, because this is what happened with CMAM, it’s gone from weight for height to weight for age and then it is three MUAC cut offs now it’s got two, you know you can only do a certain amount of changes otherwise people get confused.”*

*ID8*



**
*Where Does it Fit: IYCF or Malnutrition?*
**


The question of whether care for small and nutritionally at-risk infants u6m falls under IYCF programming or a malnutrition treatment programming also creates some confusion: ideally there would be elements of both prevention and treatment. Participants identified that this could affect the uptake of the program, as it is harder to get funding for IYCF compared to malnutrition, but they also stated that governments are more reluctant to include it into their guidelines. Furthermore, depending on the setting, it is also a question of who is responsible. In emergencies, different individuals and organizations would be responsible for the implementation of infant u6m care, according to classification as more closely aligned with IYCF or malnutrition care. 


*“…I think that is one of the problems that there is still confusion if it falls under infant feeding or acute malnutrition because of the name cMAMI. (…) The very first version of cMAMI was referred to as community-based management of acute malnutrition, and that affords what we call CMAM or IMAM in the community and that would be maybe at sometimes a different team, if you work with government it would be the same health worker but if you are working in like in the refugee context it is slightly different programmatically…”*

*ID8*



**
*Making Training Applicable to Different Levels*
**


The major critique of the IYCF-E toolkit for emergency staff was that it was targeted at staff at higher levels and was not as well suited for field staff. Thus, adaptation was needed as the training did not incorporate *“practice on how to set up lactation techniques and this type of thing.” ID3.*

However, the content on breastfeeding was found to be “fairly robust” ID3.


**
*Transition From Theory into Practice*
**


In general, the transition from theory into practice was often regarded as a limitation due to lack of practice modules in training programs and the gap between knowledge and application. 


*“I mean helping a mum breastfeeding takes time, you have to be sitting there for 40 min, half an hour, and I think when you do the training, people think oh yeah I understand this, but the actual transition into practice it is quite a big jump there.”*

*ID2*


However, for materials focusing on inpatient care there seems to be a better balance between theory and practice. Especially with the 40h Breastfeeding Course it was mentioned that trainees are able to apply the acquired knowledge in practice and thus have the possibility to build counseling skills. 


*“…is because it builds skills, it builds skills of counsellors because it includes clinical practice, so you learn a skill and then you go out and practice. You learn a new skill you keep building them and then you go out and practice.”*

*ID7*


Some training programs, such as the ‘Neonatal Infant Feeding Training’, try to overcome this gap with supervised practice, which was considered a big strength of the training as *“it gives you a lot of time to actually go into the unit and practice with real mums, real babies,…” ID5.*


**
*Breastfeeding Support*
**


Breastfeeding is complex and highly individual, with multiple complications that can arise, starting from not having enough milk to mental health issues. Thus, as a first step, participants identified the needs of having a good breastfeeding assessment as well as supporting strategies in place. Experiences were similar across different organizations and with different training tools.


*“…like when I was looking for simple tools to do breastfeeding assessment, they weren’t really available, especially not evidence based,”*

*ID1*



*“...the whole idea of counselling is quite foreign to many of our staff in emergency settings. So because it does take quite of analytical skills, I was thinking of maybe a gap that there is of how do you do a good breastfeeding assessment and then not just doing the assessment but then the next step is okay this is the issue and this is what we are going to address.”*

*ID1*


Furthermore, interviewees identified that there is a gap in the guidance for infants that cannot be breastfeed for various reasons. They identified a particular gap in how to set up an alternative feeding program, such as wet nursing.


*“So I probably say this again one of the main thing is supporting the non-breastfeed, or even relactation so trying to get them back to breastfeeding, and then I think people see it as very complicated with for example wet nursing, (…) from my experience it is actually very challenging to implement because it is not a lot of guidance on it,”*

*ID3*


Additionally, it was emphasized, there is still a big gap regarding relactation and its management, as it requires a wide spectrum of skills, ranging from technical knowledge to empathy, to not only understand the topic but also the mother’s need. Staff with these refined skills are often not available. Furthermore, from personal experience, the question was raised as to whether relactation works.


*“…it is very rare that I actually find skilled staff who know how to competently give relactation support, but I think as a sector as well we just assume that relactation works and I would actually just throw out there that I have a lot of doubts as to where lactation protocols work.”*

*ID7*



*“I think we have a tendency to focus on breastfeeding just general recommendation and lactation ahem but when you really dig in to the reasons why women are not succeeding in reaching their goals it comes back down to quite a lot of lactation management problems and issues or lack of support either wasn’t available or did not work (…).”*

*ID7*



**
*cMAMI Training Package and Implementation Guide*
**


As at the time this was only a toolkit, some participants expressed the need for an accompanying training package and a guide on how to implement it as a program. Interviewees said they had to *“learn by doing” (ID2)* and materials had to be adapted to the specific needs in the settings. Moreover, having a guide for all of the logistics that need to be considered was identified as useful, as this currently takes up time that could be focused on counseling. Moreover, it was mentioned that to implement such a training with little guidance, highly qualified staff are essential.


*“I think you need quite highly qualified staff to be able to give that training (…). So, I think my biggest kind of need, but I am not sure that is training per se related, is just a greater clarification and more guidance on how to implement.”*

*ID2*



**
*Mental Health Support*
**


Mental health support seems to be applied more in the community, and it was mentioned that in hospitals there is no “routine screening for mental health problems” (ID 4).

Even in settings where mental health is already included in the curriculum, there are still a lot of uncertainties and gaps to consider. Firstly, especially in emergency settings, staff often do not have the capacity to provide appropriate mental health support. Additionally, mental health is a complex topic that needs to be addressed by skilled staff. Thus, the question was raised on how to train staff who do not have any previous experience in this field. 


*“I just don’t feel we are quite at that point yet where we know how to do it effectively, because I think we are having the right conversations and we are getting the right information together of how to assess and how to report the mothers and if she has mental health problems but I think it is just on the ground where it is not quite working.”*

*ID3*


Different organizations seem to have different approaches in this area, as one participant mentioned:


*“(…) any other NGO, they don’t have that (…)”, but “Action against Hunger they have mental health and IYCF together, so they have quite a good position that they do that really well (…)*

*(ID3).*


Thus, bringing expertise and experience together might be useful for advancing the provision of mental health support.


**
*Practice in Providing Support*
**


Participants voiced a gap in giving trainees the chance to apply and practice newly acquired knowledge, and thus to properly consolidate and refine their skills, in supporting mothers. Training programs were often found to be too theoretical, making it hard for trainees to *“transition from abstract training, to how to apply it.”*

*(ID2).*


Participants emphasized that giving support to a mother is more than just providing recommendations; it needs experience and following certain principles that a counselor must apply, which often can only be learned when applied in practice.


*“But you know, it’s only when they actually to talk to mothers or even in the role play that they realise that they are not using the word they should. Like not using judgmental words, but it comes to them and they have (...) and stop themselves from using it, so they only do it if somebody is observing them.”*

*ID9*



**
*Mentoring, Follow-Up Training*
**


Although training in clinics was sometimes ongoing, in community settings follow up or mentoring is found to be difficult. Staff often come from a variety of different backgrounds and this knowledge is often new to many of them. Thus, mentoring and refresher trainings were thought to be essential. Furthermore, especially in emergency settings, people might encounter issues that were not taught in training programs; thus, mentoring can be useful to support staff in problem solving.


*“Having ongoing mentorship is also really important to be able to follow up and give people support outside of the classroom so I think that should really be considered during implementation.”*

*ID5*



**
*Emergency Preparedness*
**


In emergency settings providing training was experienced as more challenging, as training programs had to be compromised more and staff had to be taken away from the field during the training period. This was thought to be due to time limitations but it was also thought of as an identified gap in emergency preparedness.


*“I would say the problems with trainings is that they are given in a very rushed manner when disaster strikes and that is probably not the best, because this is really prevention, with the under six months it should be a preventive measure and it should be in areas which are prone to disasters. For instance, if you have now the Syrian refugees coming to so many countries and people know that they will be coming, so staff should be trained there on the ground to be trained of how to help mothers in how to help them to continue breastfeeding.”*

*ID9*


#### 3.2.6. Areas for Improvement: What Would Ideal Training/Support Packages Be Like?


**
*Comprehensive*
**


Regarding the question of what an ideal training package should look like, it was mentioned that the training tool should be comprehensive so that it is easy to understand, not too complicated to use, and should not add too much to field staff’s workload.


*“So again, it is about making the tool comprehensive (…) if you give someone a tool that takes them an hour to read, to document everything and they got 50 patients to see, and one takes an hour, you cannot do it.”*

*ID2*



**
*Integrated*
**


Participants suggested that instead of constantly creating new tools and guidance, programs and training should be integrated into the material that is already available. Furthermore, it was suggested that this should not only cover one sector but combine important interventions and create a curriculum that combines different sectors, such as nutrition and WASH.


*“I don’t think there is a need of new training packages but maybe to go into detail, to some of the topics or modules, that maybe need to be more extended (…)”*

*ID6*



*“So, to me it is at some point all these things have to come together. You got the early child development, you got to stimulate the child, you got baby WASH, you got to then look at the environment and separate from animals, you got the cMAMI you got to really look at breastfeeding practices and look at the status of the mother. If we think of the end point, of getting a health worker and the health systems delivering this, how can they deliver all of these different packages.*

*ID9*


This was not only identified to be important to make training more comprehensive, but also because “*topics are often overarching sectors.”* ID8


**
*Delivery of Training*
**


Participants mentioned different aspects that were regarded as ideal when it comes to the delivery of a training course. One interviewee mentioned that having a training team consisting of three people with different experience levels is helpful, as people with less experience have the possibility to gain confidence in teaching and have support when needed.


*“You know as a range of expertise in the trainer team so you have someone who is an expert and someone who is really quite new to it, but letting them run and be involved in a few trainings and build their confidence and while it is on and if they don’t know the answer, the expert trainer can answer for them, I think that works really well.”*

*ID2*


Furthermore, it was advised that training should include a variety of teaching skills. 


*“Just engaging with the materials in lots of different ways, role plays, drill like asking fast questions, sitting and thinking about it, looking at the paper. IMCI does it well, they do lots of different ways, looking at photographs and I think there are lots of different learning styles in a room, so you can kind of compensate all of it and that makes them more comfortable, rather than being forced into a learning style they are not comfortable with.”*

*ID2*


However, aligned with the previously outlined gaps, most emphasis was given to practical application of learned knowledge. Most interviewees commented that an ideal training program should include much more practical work than training programs currently involve. 


*“But then I would personally say for things like counselling and mental health it should be like a very practical workshop type. So, I think really technique content needs to be covered for a couple of days and everything else should be very, very practical.”*

*ID3*



**
*Context-Specific*
**


Settings and challenges faced are often very different and therefore training programs should be made specific to the staff who will be trained and the context they are learning in.


*“One has to prioritise so whoever who is there really needs to know what the setting is like, what the staff who are dealing with these people, what information they already had, what are the other NGOs that are already there, what information, what support they are giving?”*

*ID9*


## 4. Discussion

Ever since the WHO first recognized infants u6m as a group worthy of special focus and attention in 2013, much progress has been made. Although the importance of well-trained healthcare workers in properly supporting mothers and at-risk infants has long been known, evidence of the effectiveness of training programs and care pathways themselves is still limited [43]. Fourteen training programs/management tools were identified during the literature review. However, only seven were reported to be in regular use by our key informants.

Most training programs were found to have a similar curriculum, with variations in the content and depth by which breastfeeding, anthropometric assessment, and assessment of mothers and their mental health and counseling skills were covered. In practice, participants often drew on more than one training tool to prepare training programs specific to a setting. Thus, although training tools in theory may differ in their content, in practice the coverage might be similar due to a trainer’s expertise and emphasis on different topics.

The main issues and gaps raised by participants were regarding counseling skills for breastfeeding and mental health, ongoing monitoring, and follow-up trainings, as well as emergency preparedness. The provision of good training starts with implementation. Participants raised various issues regarding the implementation of the cMAMI toolkit, the only package to specifically focus on our target group of small and nutritionally at-risk infants u6m [36]. (N.B., since this study was conducted a new version (v3.0) of the cMAMI Tool (revamped as an integrated MAMI Care Pathway) has been released [42]). A lack of implementation guidance was reported to result in challenges regarding how to implement the toolkit. Persistent limited evidence on assessment and referral criteria remained an issue in decision making regarding when to enroll infants or when to refer to higher level health care facilities. This lack of guidance on assessment criteria becomes apparent in the literature search, as only few training programs covered anthropometric assessment in their curriculum. The importance of anthropometric assessment in the early detection of malnutrition is also reflected in the new updated WHO guidelines on the prevention and management of wasting and nutritional oedema (acute malnutrition) in infants and children under 5 years, hence, expanding curricula in this area is essential for effective detection and treatment of malnutrition [3]. Furthermore, there was uncertainty regarding whether the tool fits into an IYCF or malnutrition program, with implications for scale potential (the former attracting less funding); this may reflect the respondents’ expectation of a standardized approach. However, this will take different shapes and forms in practice. It is also likely that a closer alignment with health rather than nutrition is necessary, given that the underlying causes of malnutrition in this age group are complex and that meeting health needs are critical [44]. Lack of clarity on implementation approaches might affect the uptake of WHO recommendations into national guidelines, as programs may be more likely to be translated into policy when they are clear and based on strong underlying evidence [45,46]. This underlying strength of evidence is not yet there for global wasting guidelines [3]. 

The results of this study imply that training may be affected by a variety of factors, such as resources including time, availability, and expertise of staff, as well as the settings it is delivered in. Furthermore, a lack of time was reported to result in shortcomings in getting sufficient practical elements into training courses: more applied, practical sessions were seen as essential in training staff, equipping them with solid counseling skills, and equipping them with the skills to ensure the transition of guidelines from theory into practice. Interviews made it clear that good counseling requires practice as *“knowledge can be transmitted; however, counselling skills need to be learned in practice” (ID2).* Counseling is a subject identified as remaining foreign to many health workers. As all interviewees raised lack of time and practice as an issue, it is likely that this might hinder HW in developing optimal counseling skills. 

A study investigating the effectiveness of the WHO 40 h Breastfeeding Counseling course came to similar conclusions [47]. The predominant challenge in covering content was time, resulting in the shortening of clinical practice and exercises. In concordance with the current study, good counseling skills were not only regarded as theoretical knowledge but rather as soft skills, such as good communication, listening, and supporting skills, as well as a high level of empathy, essential in the provision of breastfeeding support.

Especially in emergency and low-income settings, health facilities must often draw on staff dependent on availability rather than their expertise. Participants identified that staff may have different backgrounds and previous levels of knowledge and competencies, making it more difficult to provide training that meets each individual’s needs. A solution proposed to help address this was preparation, as it is important for a trainer to know a setting and participants prior to a training program’s commencement.

Considering both time and staff availability, it becomes apparent that these are interlinked in their influence on training outcomes and may have an adverse effect on the quality of service. Especially with the goal in mind of shifting towards a more holistic approach in managing mothers and nutritionally at-risk infants u6m, and of addressing mothers’ mental health, good counseling skills are key. However, participants raised the concern that the implementation of mental health counseling in practice has its own difficulties. Often, neither facilitators nor trainees have previous experience in this field, making it difficult to provide training that immediately translates into quality service. Only four training programs directly covered mental health counseling, of which ‘Baby Friendly Spaces’ by AFC was mentioned as having a particularly good approach [39]. This review of the training confirmed that it follows a practical and comprehensive approach to ensure good psychological support. It builds on a clear guide and on how to implement and provide mental health support based on ‘IASC psychosocial guidelines’.

Finally, the need to monitor and evaluate training programs was identified. The current gap in this regard is reflected in the finding that only six training programs dedicated a chapter to monitoring and evaluation. Only two studies evaluating the effectiveness of training programs were found [47,48]. Our key informants emphasized that not only provision of training programs should be evaluated, but also the degree to which this results (or not) in good quality services. Monitoring and evaluation indicators are needed. ‘Baby Friendly Spaces’ was the only training program providing a variety of example indicators.


**Areas for Improvement**


To ensure the good quality and effectiveness of training programs, different themes were raised during the interviews: training programs must be comprehensive, integrated, include different delivery methods, and be context-specific. It was emphasized that training programs cannot be provided as one-off events and that follow-up training programs are essential. However, this only seemed to be the norm in inpatient settings. A lack of refresher training programs was outlined in emergencies, as response must be fast, and trainers are often only available for short time. A study evaluating humanitarian response concluded the same, and that follow-up training programs are a must in the assurance of quality [49].

Concerns about comprehensiveness were raised in particular regarding the cMAMI tool. It was perceived as too complicated, and participants mentioned that even staff with a nutrition background struggled with its application. However, given that the tool is modeled on the WHO-developed Integrated Management of Child Illness (IMCI), it may be more accessible to those with a health rather than nutrition background. Furthermore, this tool has since been significantly updated and reframed based on an integrated care pathway model, which may serve to bring greater clarity on how it connects and is located within core health and nutrition guidelines and protocols. 

Finally, it was suggested that training programs should employ a variety of delivery methods, such as lectures, visuals, exercise, group simulations, and clinical practice. As healthcare workers often have different backgrounds, and individuals generally have different learning strategies, this means that different ways to deliver core knowledge and develop competencies are needed. Different approaches suit different people and thus a range of options makes a course as applicable as possible to as many as possible. Therefore, it has greater chances of having a future impact on the provision of care. 


**Strengths and Limitations**


Rather than evaluating the impact of training programs specifically, studies often focus on the effectiveness of interventions when delivered in tightly controlled trial settings, which are rarely representative of real-life situations. In contrast, this study not only summarizes training programs available and in use by different organisations, but it also provides an idea of their practical application and areas that still need improvement. Due to the variety of participants’ backgrounds, the study was able to provide an insight into the application of training programs in different fields and identify a broad spectrum of challenges. 

We also, however, acknowledge limitations: 

Firstly, this was not an exhaustive systematic review, and it is likely that there are more training programs and support tools available that we did not identify. 

Secondly, most participants for our key informant interviews were identified via the MAMI Global Network and therefore their experience cannot necessarily be reflective of all people working in this field globally. In addition, those interviewed were largely engaged with international agencies and those working at national and sub-national level in training were not represented. This will give a particular perspective to our findings that is important to consider when interpreting. Our recruitment strategy via the MAMI Global Network may have also heightened the capacity to critique the cMAMI tool (developed by this global initiative), potentially narrowing the identification of challenges faced by other training programs. As noted earlier, the cMAMI tool discussed in this paper has since been updated to version 3; this is likely better than its predecessor, though data to confirm this are needed. 

Thirdly, we do not know how our tools/materials nest within and complement (or not) wider health and nutrition policies. This is a very setting-specific issue, which we hope that future researchers will address.

Finally, the study did not include a key informant who had experience in using the ACF training tool. As some participants mentioned that this training tool had a good approach in tackling mental health counseling, more insight into this training tool would have been of particular interest.


**Future Research and Development**


To improve the delivery of future training programs, we suggest: Rather than developing completely new additional trainings, future research should focus on how to effectively integrate care of small and nutritionally at-risk infants u6m into future editions of existing training programs available across health and nutrition. The update to Integrated Management of Childhood Illness currently underway by WHO provides one critical opportunity to do so.More evidence is needed on how, where, and in what way the training packages we flagged feed into developing heath policies, including policies on breastfeeding support.As breastfeeding and relactation support needs are varied and may be complex, more specific training on how to provide this is needed at all stages of a healthcare worker’s career (i.e., undergraduate, postgraduate in-service training, postgraduate formal, and specialist training). Training should include skills in how to effectively re-establish exclusive breastfeeding in at-risk infants u6m and growth outcomes.More evidence is needed regarding how to provide health workers without psychological backgrounds with the skills to assess and provide effective mental health counseling.Evaluation of the effect of training programs on the quality of services is needed, as well as strategies to sustain quality care provision.Training programs should incorporate context-specific practical implementation; the complexity that will limit educational approaches on how best to achieve this requires exploration.

## 5. Conclusions

In this review, we identified a number of relevant training packages. All were useful and covered some relevant topics, but no one package alone fully and directly met the requirements to train and develop staff caring for small and nutritionally at-risk infants u6m. 

Limiting factors to effective training package delivery included time, staff availability, consideration of the context in which trainings are held, and previous experience doing the work. The biggest limitation is that current training packages focus mainly on narrow, stand-alone topics. What is needed for future care is a more holistic package that uses elements of these stand-alone tools but brings them together for a more integrated and thus more effective, coherent whole. It would be useful to have a core curriculum and package endorsed by a respected and authoritative international organization. However, any such package should have options for adaptations and local adjustments according to local patterns of risk factors underlying infant u6m malnutrition: local knowledge, local preferences, and users’ specific training needs. Specific implementation research on the effectiveness of training would further the value and impact of future training packages. 

## Figures and Tables

**Figure 1 children-10-01496-f001:**
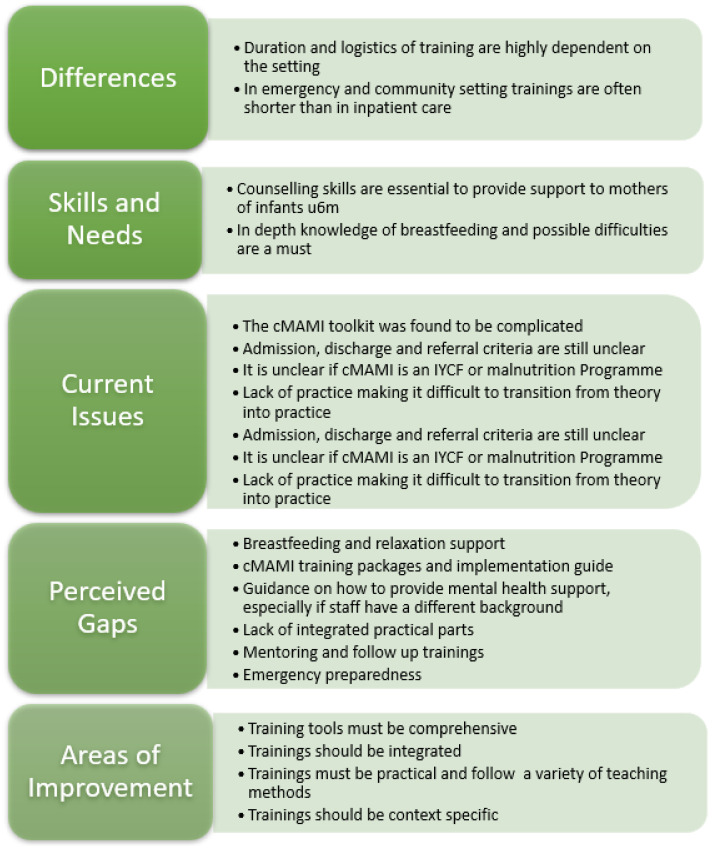
Coded themes and subthemes arising during key informant interviews.

**Table 1 children-10-01496-t001:** Training packages and patient support materials relevant to small, nutritionally at-risk infants u6m.

	A	B	C	D	E	F	G	H	I	J	K	L	M	N
Curriculum Covered	Harmonised Training Package (HTP) [28]	Breastfeeding Counselling a Training Course [29]	Community IYCF Counselling Pack [30]	IYCF Counselling: an Integrated Course [31]	IMCI [32]	Infant Feeding in Emergencies “Module 2” [33]	HIV Infant Feeding Counselling [34]	Think Healthy [35]	cMAMI Tool (v.2) [36]	Getting to Know Cerebral Palsy [37]	IYCF-E Toolkit [38]	Baby Friendly Spaces [39]	Feeding + Positioning Manual [40]	IYCF Counselling Community-Focused Approach [41]
**Infant**														
Feeding Assessment	X	X	X	X	X	X			X	X	X	X		
Anthropometry/Nutritional Assessment					X				X			X		X
The Sick Child		X			X									X
LBW Babies		X	X	X	X	X			X		X	X	X	X
Breastfeeding	X	X	X	X	X	X	X		X	X	X	X	X	X
Relactation		X				X			X		X	X		
**Mother**														
Health Assessment								X	X		X	X		X
Nutrition, Physical Health		X	X		X	X		X	X		X	X	X	X
HIV and Infant Feeding		X	X	X		X	X		X		X	X	X	X
Disability										X			X	
**Counselors**														
Counseling Skills		X	X	X	X	X	X	X		X	X	X		X
Communication Skills		X	X	X	X	X		X			X	X		X
**Other**														
Management of Artificial Feeding/Donations	X			X		X					X	X		
Emergency Preparedness	X					X					X	X		
Integration into Other Nutrition Programs				X					X		X	X		
Food Hygiene		X				X					X	X		

**Table 2 children-10-01496-t002:** Key informant characteristics.

Participant ID	Organization Type	Setting
1	NGO	Outpatient Care
2	Academic/Research Institution	Inpatient Care
3	NGO	Emergencies
4	Hospital	Inpatient Care
5	NGO	In- and Outpatient Care
6	NGO	Community Setting
7	NGO	Inpatient Care
8	NGO	Community Setting
9	Nutrition Initiative	Emergencies

**Table 3 children-10-01496-t003:** Logistics of training programs and patient support tools as described by key informants.

ID	Training Used	Setting	Target Audience	Duration	Costs	Curriculum	Guidelines Followed
1	cMAMI	Emergencies, communities	IYCF counselors, nurses	2 days on cMAMI, 3-day IYCF plus 2 days and then once a week	No direct costs or travel costs for staff	Save the Children IYCF training	Modified from cMAMI guidance, national guidelines, WHO
2	IMCI, IYCF, cMAMI	Hospital, communities	CHW, IMCI: doctors, nurses, paramedics	3 to 5 days, IMCI and IYCF often running for 3 to 4 months in inpatient	N/A	IMCI curriculum, IYCF curriculum, breastfeeding	IYCF, FANTA
3	IYCF, IYCF-E, cMAMI	Emergencies	IYCF counselors, managers, trainers	5 days for managers, for counselors usually 3 days	Venue, food during trainings, material, baby dolls, relactation kit	IYCF-E curriculum, relactation, exclusive BF, topics related to <6 months, counseling skills, setting up an artificial feeding program, psychosocial support, staffing	WHO, UNICEF
4	Tailor-Made Course	Hospital	HW, doctors, nurses, nutritionists, peer supporters	5 days	N/A	Pathology of malnutrition, clinical management of malnutrition	Baby-friendly hospital initiative, national guidelines for management of acute malnutrition
5	Neonatal Feeding Training	Inpatient clinic	Nurses, social workers/HW with different backgrounds, midwives	5 days	N/A	Assessment, diagnosis, treatment, what is good BF: how many times a day, breastfeeding assessment, counseling, monitoring	National guidelines, malnutrition guidelines from WHO, UNICEF, FANTA
6	Community IYCF Counseling Pack	Community	Nutrition officers, protection officers, community mobilisers, nutrition coordinators	3 to 5 days	N/A	Community awareness, community sensitization, content for under six months	WHO, UNICEF
7	40h Breastfeeding Course, Tailor-Made Course, Infants With Feeding Difficulties	Inpatient	Front-line community workers, healthcare staff	5 days	N/A		
8	cMAMI	Emergencies	Nurses with nutrition experience	1 to 3 days	N/A		cMAMI toolkit, national guidelines
9	Training Based on WHO, UNICEF and Later Modified Version, IYCF-E	Emergencies	Variety off people with different backgrounds	Depending on setting and resources	N/A	Growth monitoring, taking anthropometrics	

## Data Availability

Data is available from the study authors on request.

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
