# Peer review of "Training Packages and Patient Management Tools for Healthcare Staff Working with Small, Nutritionally At-Risk Infants Aged under 6 Months: A Mixed-Methods Study†"

_children, 2023, doi:10.3390/children10091496_

Round 1

Reviewer 1 Report

Dear Editor,

Thank you for providing the opportunity to review the article titled "Training packages and patient management tools for healthcare staff working with small, nutritionally at-risk and malnourished infants aged under 6 months: a mixed-methods study".

I think that this study will contribute to the public health studies on the prevention of malnutrition in infants.

When comparing the features of the training packages examined in this study with each other, I think it would be useful to state whether they cover the following topics:

Risky babies discharged from the neonatal intensive care unit (premature, babies with congenital anomalies, undergone surgery, etc.)

Practices and measures that can enable early diagnosis within the standard baby follow-up system

Small for gastational age babies

Developing health policies for the dissemination of breastfeeding support and relaxation units

Providing breastfeeding support trainings before and after graduation to primary healthcare providers  and pediatricians working in primary health care services.

Establishing health policies for the establishment of mechanisms to control whether infants have access to primary health care services and to determine access to health services.

Providing access to outpatient and inpatient health services for babies with malnutrition and preventing the recurrence of malnutrition

Ability to make a dynamic and individual approach according to regions or societies

Prevention of malnutrition in mothers

Tables should be easier to understand and easier to read.

I believe that it would be useful for this article to be evaluated by experienced community pediatricians and public health specialists.

Sincerely.

Author Response

Dear Dr Jin, Children Editors & Reviewers,

Re: "Training packages and patient management tools for healthcare staff working with small, nutritionally at-risk infants aged under 6 months: a mixed-methods study".

Thank you for this opportunity to resubmit minor corrections to our article. We are grateful for your constructive suggestions and for giving us the opportunity to further improve our work.

Especially with the newly launched July 2023 WHO guidelines for infant and child wasting (malnutrition), our piece:

-  is more topical and newsworthy than ever.

-  has an important role to play in informing future policy, practice and infant malnutrition treatment programme rollout.

- will, we believe, be of great interest and relevance to the wide international readership of CHILDREN journal (and others who find the work on Pubmed and other search engines)

- will be well cited due to the above.  

We respond to your comments and suggestions point by point below.

Please also note, as per email correspondence with editor Dr Jenny Jin,, we have also added a further reference to the original MSc thesis project on which this was based_

https://discover.lshtm.ac.uk/permalink/44HYG_INST/od12o/alma991000807713103736

We have also thoroughly proofread and done further minor typographical corrections in the text. We trust it reads better as a result.

We thank you for your kindness and understanding and hope that these minor corrections now meet your full criteria for acceptance

With thanks and kind regards,

Dr Marko Kerac

(Corresponding author, on behalf of co-authors)

REVIEWER 1

Thank you for providing the opportunity to review the article titled "Training packages and patient management tools for healthcare staff working with small, nutritionally at-risk and malnourished infants aged under 6 months: a mixed-methods study".

I think that this study will contribute to the public health studies on the prevention of malnutrition in infants.

OUR REPLY #1:

Thank you for your encouragement and positive feedback about the value of our study.

We trust that CHILDREN journal readers and other researchers will also share your opinion and appreciate the relevance of this article in the same way. Your time to review our study is sincerely appreciated and we are thankful for your detailed suggestions to enhance the quality of the article.

When comparing the features of the training packages examined in this study with each other, I think it would be useful to state whether they cover the following topics:

Risky babies discharged from the neonatal intensive care unit (premature, babies with congenital anomalies, undergone surgery, etc.)

OUR REPLY #2:

Thank you very much for this valuable suggestion. Our guidelines are indeed intended to apply to this group. We have:

- further clarified who is our target population on page 2, para 2, lines 49-66

- also combined this suggestion with the following “Small for gestational age babies” and included a column in Table 1. We named the column “LBW and sick babies”.

Practices and measures that can enable early diagnosis within the standard baby follow-up system

OUR REPLY #3:

Thank you very much for this suggestion. Indeed, only view trainings cover detailed instructions on how to conduct anthropometric assessments, which is an important tool for early detection and diagnosis of malnutrition. We therefore included this in the discussion as limitations of trainings (see line 606-612).

Small for gestational age babies

OUR REPLY #4:

Thank you again for this point. Now included and described this group in reply #2

Developing health policies for the dissemination of breastfeeding support and relaxation units

OUR REPLY #5:

Developing health policies for the dissemination of breastfeeding support and relaxation units is indeed very important when supporting small and nutritionally at-risk infants aged u6m. The extent to which these training packages link with and feed into policy is slightly beyond the scope of this work and it is not possible for us to say on the basis of our qualitative data how often this actually occurred, not least because it is a very setting-specific issue. However, we:

- Have now recognized this important issue in the background in lines 89-92.

- Hope that our work will inspire other researchers to do this work looking at the link – thus we make this point bullet point #2 in the ‘future research & development” section.

- We also acknowledge the issue in point number three of our limitations section.

Providing breastfeeding support trainings before and after graduation to primary healthcare providers and pediatricians working in primary health care services.

OUR REPLY #6:

We combined this response with the one provided under reply 5. We:

- Mention this need in line 89-92

- Highlight the need to do more on this in the “future research & development” section, bullet point #3.

Establishing health policies for the establishment of mechanisms to control whether infants have access to primary health care services and to determine access to health services.

OUR REPLY #7:

As with point #5, this important issue of health policies is beyond the scope of this current work since we are just focusing on training and support materials rather than how these may/may not be used in large scale health policies. However, by flagging the issue in edits and expansions previously noted, we hope that others will be able to look at this topic and thus build on our research and further build strong evidence in this area.

Providing access to outpatient and inpatient health services for babies with malnutrition and preventing the recurrence of malnutrition

OUR REPLY #8:

Trainings have been classified according to the settings in which they are see lines 250-253 e.g. whether they are aimed at staff working in in- or outpatient settings. To include this suggestion, we have also included a row in table 3 outlining coverage in curriculum of how to integrate trainings into other nutrition interventions. It has also been mentioned in line 272-274. We appreciate this suggestion, as it is in line with participants concerns of making trainings integrated.

Ability to make a dynamic and individual approach according to regions or societies.

OUR REPLY #9:

This is a very good point but is also very much a function of the team/organization delivering training rather than inherent to individual packages themselves.

Trying our best to capture the issue of how a individual trained healthcare worker can best make a dynamic and individual approach we did identify that excellent counselling skills are seen as a key skill here. Hence, we compared how much emphasis various courses put on these counseling skills – please see line 269-271.

Prevention of malnutrition in mothers

OUR REPLY #10:

Thank you for this feedback, as we already compared maternal health, mental health and malnutrition, we now expanded on topics covered regarding maternal nutrition/malnutrition after your feedback in line 257-263.

Tables should be easier to understand and easier to read.

OUR REPLY #11:

Thank you very much for this valuable feedback. It is very important to us to make the content as easy to understand as possible and hence we adjusted the layout of the tables as much as we could while adhering to the journals formatting. If further changes are needed we appreciate the kind input of the journals’ professional type-editing staff who we are certain can make the tables even slicker and better.

I believe that it would be useful for this article to be evaluated by experienced community pediatricians and public health specialists.

OUR REPLY #12:

Thank you again for your support. We indeed hope our future readers will include these groups.

Reviewer 2 Report

The paper is very interesting and contains relevant information.

The conclusions should be completely rewritten

DOI is missing

Table 1 to improve not readable

Table 3 to be improved not readable

topic to improve

Author Response

Dear Dr Jin, Children Editors & Reviewers,

Re: "Training packages and patient management tools for healthcare staff working with small, nutritionally at-risk infants aged under 6 months: a mixed-methods study".

Thank you for this opportunity to resubmit minor corrections to our article. We are grateful for your constructive suggestions and for giving us the opportunity to further improve our work.

Especially with the newly launched July 2023 WHO guidelines for infant and child wasting (malnutrition), our piece:

-  is more topical and newsworthy than ever.

-  has an important role to play in informing future policy, practice and infant malnutrition treatment programme rollout.

- will, we believe, be of great interest and relevance to the wide international readership of CHILDREN journal (and others who find the work on Pubmed and other search engines)

- will be well cited due to the above.  

We respond to your comments and suggestions point by point below.

Please also note, as per email correspondence with editor Dr Jenny Jin,, we have also added a further reference to the original MSc thesis project on which this was based_

https://discover.lshtm.ac.uk/permalink/44HYG_INST/od12o/alma991000807713103736

We have also thoroughly proofread and done further minor typographical corrections in the text. We trust it reads better as a result.

We thank you for your kindness and understanding and hope that these minor corrections now meet your full criteria for acceptance

With thanks and kind regards,

Dr Marko Kerac

(Corresponding author, on behalf of co-authors)

REVIEWER 2

The paper is very interesting and contains relevant information.

OUR REPLY #1:

Thank you for your encouragement and positive feedback about the value of our study.

We trust that CHILDREN journal readers and other researchers will also share your opinion and appreciate the relevance of this article in the same way. Your time to review our study is sincerely appreciated and we are thankful for your detailed suggestions to enhance the quality of the article.

The conclusions should be completely rewritten

OUR REPLY #2:

We have made edits both here and throughout the manuscript and trust there is a big improvement as a result. Thank you!

DOI is missing

OUR REPLY #3:

This will presumably be added on full acceptance of our piece. We have left blank for now and will be guided by CHILDREN editorial team.

Table 1 to improve not readable

OUR REPLY #4

Thank you for this opportunity to clarify. We have made edits and trust that this is now better. If further clarity is needed, we would be grateful for any further input that the journal expert typesetters can provide to improve the appearance and readability of our tables – this expert help would be very much appreciated.

Table 3 to be improved not readable

OUR REPLY #5

As for table 1: thank you for this opportunity to clarify. We have made edits and trust that this is now better. If further clarity is needed, we would be grateful for any further input that the journal expert typesetters can provide.

topic to improve

OUR REPLY #6

Again, we thank you for overall support for this work and trust that the edits made as a result of the peer review process further strengthen our article.